# Groundwater Suitability for Drinking and Irrigation Using Water Quality Indices and Multivariate Modeling in Makkah Al-Mukarramah Province, Saudi Arabia

**Maged El Osta [1],*** , **Milad Masoud [1]** , **Abdulaziz Alqarawy [1,2]** , **Salah Elsayed [3]** and **Mohamed Gad [4]**

1   Water Research Center, King Abdulaziz University, Jeddah 21589, Saudi Arabia;
    mhmasoud@kau.edu.sa (M.M.); aalqaraawi@kau.edu.sa (A.A.)
2   Faculty of Meteorology, Environment and Arid Land Agriculture, King Abdulaziz University,
    Jeddah 21589, Saudi Arabia
3   Agricultural Engineering, Evaluation of Natural Resources Department, Environmental Studies and
    Research Institute, University of Sadat City, Sadat City 32897, Egypt; salah.emam@esri.usc.edu.eg
4   Hydrogeology, Evaluation of Natural Resources Department, Environmental Studies and Research Institute,
    University of Sadat City, Sadat City 32897, Egypt; mohamed.gad@esri.usc.edu.eg
*   Correspondence: melosta@kau.edu.sa

**Abstract:** Water shortage and quality are major issues in many places, particularly arid and semi-arid regions such as Makkah Al-Mukarramah province, Saudi Arabia. The current work was conducted to examine the geochemical mechanisms influencing the chemistry of groundwater and assess groundwater resources through several water quality indices (WQIs), GIS methods, and the partial least squares regression model (PLSR). For that, 59 groundwater wells were tested for different physical and chemical parameters using conventional analytical procedures. The results showed that the average content of ions was as follows: $Na^+ > Ca^{2+} > Mg^{2+} > K^+$ and $Cl^- > SO_4^{2-} > HCO_3^{2-} > NO_3^- > CO_3^-$. Under the stress of evaporation and saltwater intrusion associated with the reverse ion exchange process, the predominant hydrochemical facies were $Ca$-$HCO_3$, $Na$-$Cl$, mixed $Ca$-$Mg$-$Cl$-$SO_4$, and $Na$-$Ca$-$HCO_3$. The drinking water quality index (DWQI) has indicated that only 5% of the wells were categorized under good to excellent for drinking while the majority (95%) were poor to unsuitable for drinking, and required appropriate treatment. Furthermore, the irrigation water quality index (IWQI) has indicated that 45.5% of the wells were classified under high to severe restriction for agriculture, and can be utilized only for high salt tolerant plants. The majority (54.5%) were deemed moderate to no restriction for irrigation, with no toxicity concern for most plants. Agriculture indicators such as total dissolved solids (TDS), potential salinity (PS), sodium absorption ratio (SAR), and residual sodium carbonate (RSC) had mean values of 2572.30, 33.32, 4.84, and −21.14, respectively. However, the quality of the groundwater in the study area improves with increased rainfall and thus recharging the Quaternary aquifer. The PLSR models, which are based on physicochemical characteristics, have been shown to be the most efficient as alternative techniques for determining the six WQIs. For instance, the PLSR models of all IWQs had determination coefficients values of $R^2$ ranging between 0.848 and 0.999 in the Cal., and between 0.848 and 0.999 in the Val. datasets, and had model accuracy varying from 0.824 to 0.999 in the Cal., and from 0.817 to 0.989 in the Val. datasets. In conclusion, the combination of physicochemical parameters, WQIs, and multivariate modeling with statistical analysis and GIS tools is a successful and adaptable methodology that provides a comprehensive picture of groundwater quality and governing mechanisms.

**Keywords:** multivariate modeling; physicochemical parameters; water quality indices; hydrochemical facies; GIS techniques

## 1. Introduction

Groundwater is a life-sustaining and crucial resource of the planet [1] (Busico et al., 2020). Water crises and quality are major concerns in many countries, especially arid and semi-arid regions where water shortages are common, and little attention has generally been given to assessing water quality [2,3]. Arid and semi-arid regions suffer from multiple critical issues such as scarcity of water resources and extensive exploitation of groundwater for different uses [4]. These problems will certainly cause a decline in water tables and the degradation of groundwater quality. Therefore, Wadi Fatimah basin was selected for this study. It is a large basin in Makkah province with an area of about 4869 km$^2$ [5], and is considered the main source of water supply for many cities and villages.

Aquifers are especially vulnerable to the effects of uncontrolled extraction and insufficient land use, putting groundwater quality at danger [6–8]. The quality of water at these resources needs proper attention, especially since pure water is essential for drinking, agriculture, and residential use [9]. Groundwater monitoring is critical to meeting growing water needs in respect of availability and quality, and it must be implemented [10]. Recently, a substantial and expanding body of research examined water resources with a focus on evaluating and understanding hydrochemical properties and groundwater quality by utilizing a variety of effective approaches. Water quality indicators (WQIs), geographic information system (GIS) techniques, statistical methodologies, and multivariate modeling are examples of such strategies [11]. The physicochemical properties of water may be utilized to fully comprehend and identify elements influencing groundwater quality as well as to give vital information for water management. Water characteristic, which is established on physicochemical criteria, gives current information on water facies, various geochemical controlling mechanisms, and water classes [12–14]. Water chemistry and geochemical characteristics provide a good basis for examining trends, describing particular sustainability issues, and transferring knowledge on sources of water, geochemical dynamics, quality of water, and water susceptibility for drinking and irrigation [15,16].

The geochemical characteristics of groundwater are essentially governed by recharging, aquifer metrics, contact time, and specific geochemical mechanisms such as dissolution, mineral solubility, and ion exchange processes [17–20]. Therefore, water quality management should be decided by a complete groundwater quality evaluation with respect to physicochemical features and variables influencing water quality [21–24]. Consequently, derivative approaches for defining the key geochemical factors that govern water quality and analyzing the mixing process between fresh and saline water, such as Piper trilinear diagram, Chadah diagram, Gibbs diagram, and hydrochemical facies evolution diagram (HFE), are appropriate and commonly applied [25,26].

Furthermore, multivariate analyses such as cluster analysis (CA) and principal component analysis (PCA) are effective techniques for identifying key physicochemical characteristics and the interrelationships between these parameters in order to comprehend the major variables driving the distribution of physicochemical metrics in water [27–29]. In addition, finding associations between multiple physicochemical variables may be regarded as a unique step forward towards groundwater quality management using statistical correlation analysis, which has been demonstrated to be a highly suitable approach.

The WQIs are derived from a big data collection containing different water quality metrics from various places. Several WQIs were developed to serve as indicators for assessing water availability in both potable supply and agricultural usage [30–33]. The basic goal of WQIs is to convert large numbers of complicated datasets into quantitative water quality data, contributing to a better understanding of water quality [34]. The drinking water quality index (DWQI) may be developed as a reliable tool described as a value that represents the combined impact of many water quality variables [35]. Therefore, DWQI is calculated by analyzing the cumulative impact of man-made and natural activities based on certain factors in the hydro-geometric properties of the groundwater sample.

Based on experience and judgment, irrigation water quality indices (IWQIs) such as TDS, potential PS, SAR, and RSC can meet the criteria for appropriate controls and further

evaluating water validity for agricultural purposes [36–38]. For example, the IWQI is a significant and distinct model of these indices utilized in water quality evaluation and agricultural production optimization [39,40].

Based on a variety of physicochemical parameters, statistical approaches, WQIs, geographic information system (GIS) techniques, and multivariate modeling are employed [30,41–43]. Utilization of physicochemical parameters, WQIs, and multivariate modelling with statistical analysis and GIS tools is a successful and practical strategy that provides a complete picture of water quality and governing mechanisms.

Determination of DWQI and IWQI requires a long sequence of calculations to convert multiple numbers from the physicochemical elements data into a single value that reveals the validity of water quality level for drinking and irrigation usage. In order to overcome this problem, PLSR approaches were applied in this work. The PLSR method can be used to select the most effective parameters for calculating the DWQI and IWQI. This, in turn, leads to a reduction in the number of elements utilized in the chemical analysis to determine WQIs as well as a reduction in cost. The PLSR approach is a popular way to describe a linear relationship between independent and dependent parameters [44]. PLSR can reduce many collinear components to a few non-correlated latent variables, decreasing duplicate data and minimizing overfitting or underfitting [45,46]. Based on the advantages of these methods, the WQIs can be simultaneously computed from numerous big data using these approaches.

For that, the goals of this research were to (i) identify hydrochemical facies and geochemical processes using physicochemical metrics; (ii) investigate the geochemical controlling factors influencing the chemistry of water using imitative techniques; (iii) evaluate the appropriateness of groundwater for drinking and irrigation with respect to WQIs; and (iv) evaluate the performance of PLSR models based on investigated physicochemical parameters in forecasting the six WQIs, namely DWQI, IWQI, TDS, SAR, PS, and RSC.

## 2. Research Materials and Methods

### 2.1. Area of Study and Description

Makkah Al-Mukarramah province is considered as one of the most important regions in the KSA because of its religious and historical standing, and the large number of residents and tourists who come visiting. The administrative territory of the Makkah region is bordered in the west by the Red Sea, in the east by Riyadh, in the north by Al-Madinah Al-Munawarah, and in the south by the provinces of Al Bahaand Asir, with an area of about 141,216 km$^2$ (Figure 1b). Topographically, Makkah region is characterized by a great diversity in altitude between 0 m and 2984 m above sea level (amsl), while the lithological units that dominate this area belong to pre-Cambrian to Quaternary age (Figure 2).

The Precambrian rocks are mainly late-Proterozoic basaltic to rhyolitic volcanic, volcano-clastic, and epi-clastic rocks of the basic island-arc type that have been repeatedly distorted and metamorphosed by intrusion rocks of various ages and mixtures. Below a layer of horizontally basaltic lava and Quaternary sediments, the Tertiary stratigraphic sequence is apparent. It is primarily made up of clastic rocks that are dominated by sandstone, shale, mudstone, oolitic ironstone, and conglomerates. The average annual precipitation in this region varies between 50 mm/year and 400 mm/year with some shifts in dry and wet years. The amount of rain falling on this area represents the main source for recharging the groundwater aquifers. Accordingly, this district contains many Wadies (44 catchments) extending to the north and east, which are characterized by surface and groundwater resources [47]. Therefore, scientific research and projects must be directed to this strategic region in order to sustain its water resources and achieve the Kingdom's "Vision 2030". The present study focuses on Wadi Fatimah basin, which spans a broad area of the south and east part of the Jeddah governorate and extends from NE to SW with about 4869-km$^2$ area. It is located between the longitudes 39°15′ and 40°30′ E and latitudes 21°16′ and 22°15′ N (Figure 1c). The Quaternary aquifer is the primary source of groundwater for different uses in this Wadi. This aquifer is primarily made up of conglomerates, sandstone,

and mudstone (Wadi fill deposits) that range in thickness from 10 to 60 m based on the data of drilled wells in the area. The igneous metamorphic rocks that make up the bedrock of this aquifer are highly fractured and weather-cracked, making them the perfect host for groundwater preservation. The depth of groundwater varies from 1.2 to 50.1 m from the ground surface, with an average value of 16.7 m according to the field measurements from 64 drilled wells. Based on this data, a water-level distribution map was constructed to show the flow of groundwater along the Wadi. As shown in Figure 3, the groundwater flow was from the east to west direction towards the Red Sea.

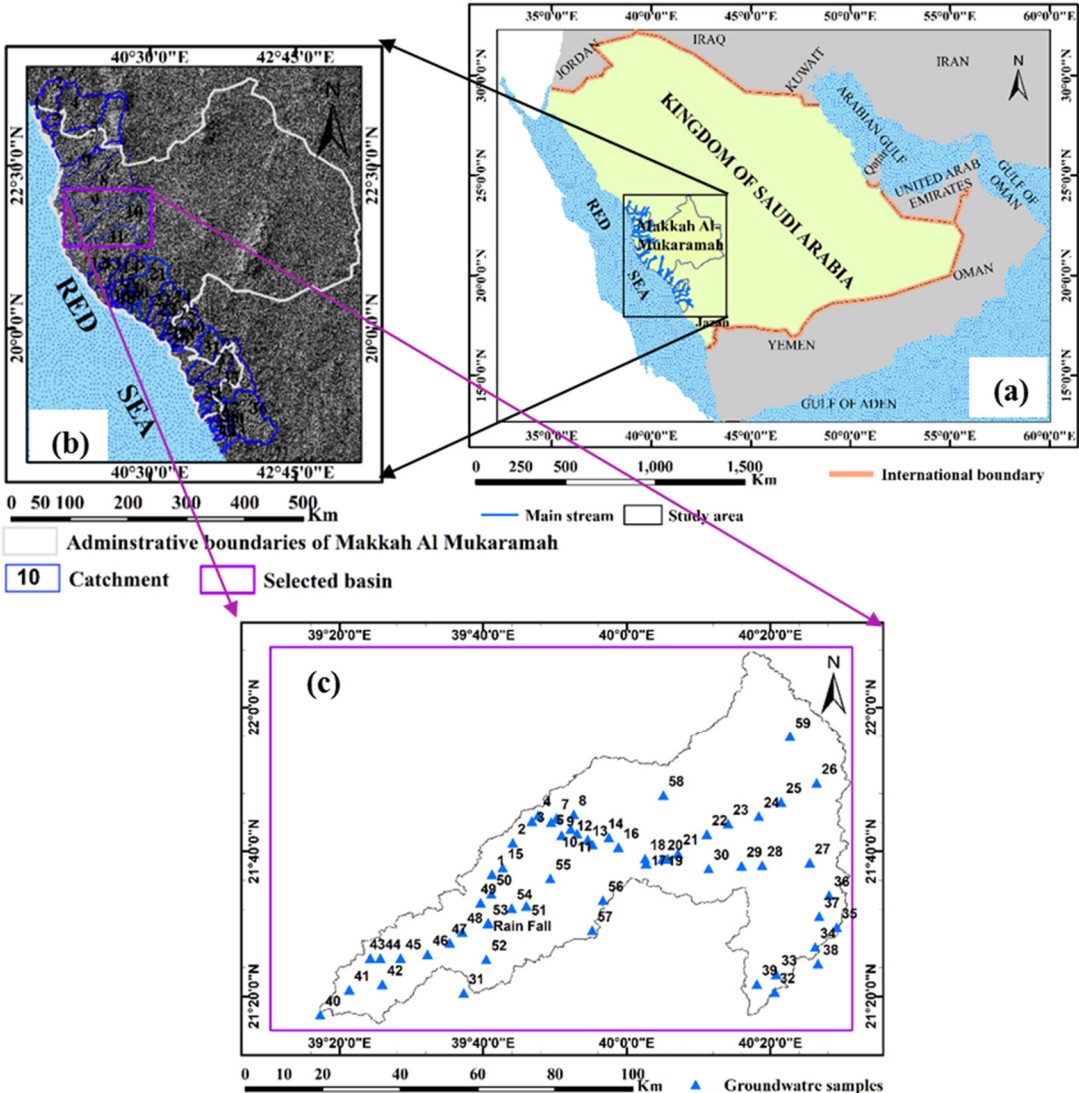

**Figure 1.** (**a**) Location map of Makkah Al-Mukarramah province, (**b**) Wadi Fatimah basin and (**c**) locations of groundwater samples.

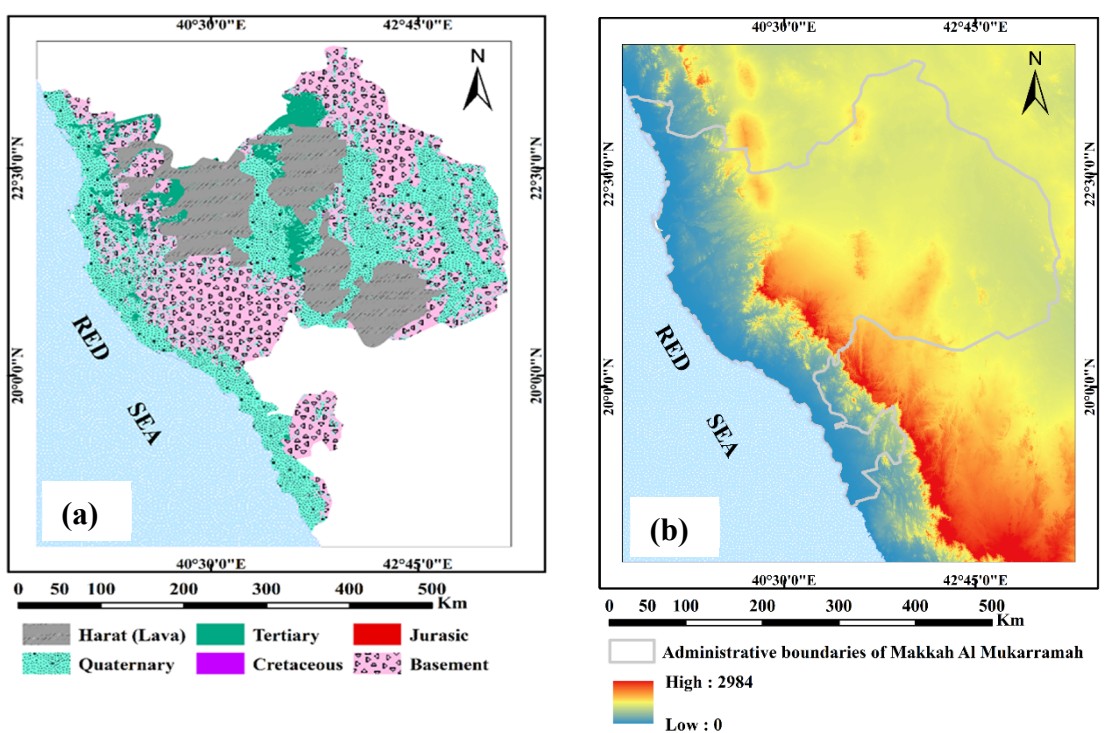

**Figure 2.** (**a**) Geological map of Makkah Al-Mukarramah province, and (**b**) digital elevation model (DEM).

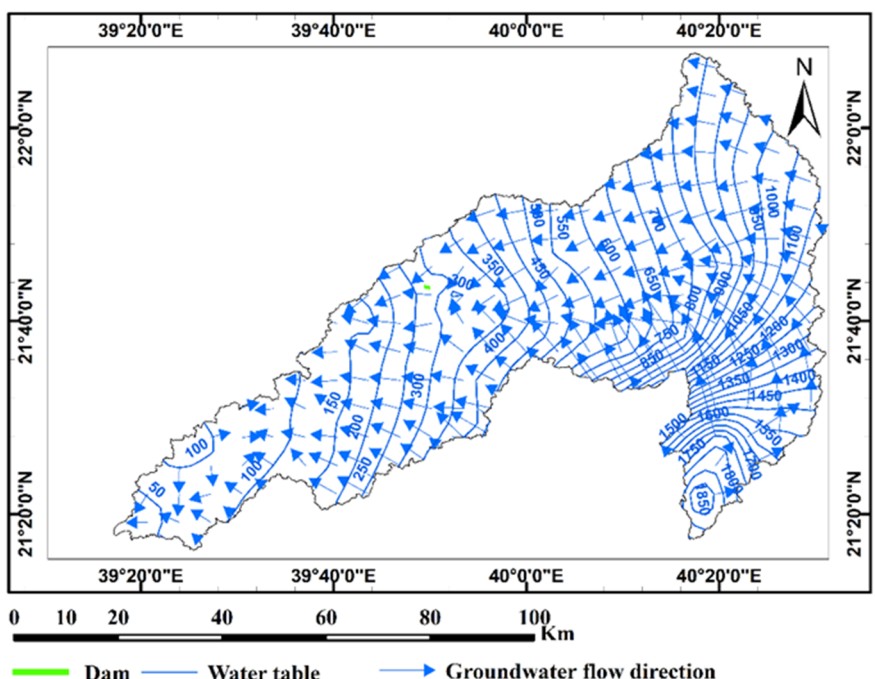

**Figure 3.** Groundwater elevation contour map in meter (above msl) and flow direction in the area.

### 2.2. Samples Collection and Analytical Procedures

During the year of 2021, 59 samples of groundwater were obtained from the Quaternary aquifer in Wadi Fatimah basin, Makkah Al-Mukarramah province, KSA, for estimating water quality for both drinking and irrigation. Portable Magellan GPS 315 was used to detect the water samples position and identify UTM coordinates of the study area (Figure 1c). The water samples were stored at 4 °C until they were taken to the laboratory for physicochemical analysis examination. Thirty different physicochemical parameters were detected by standard analytical methods [48]. A portable conductivity multi-parameter apparatus

was used to measure temperature, pH, EC, and TDS at the site (Hanna HI 9033), calibrated with standard solutions. $Mg^{2+}$ and $Ca^{2+}$ concentrations were determined by the EDTA titrimetric technique using ethylenediaminetetraacetic acid, whereas $K^+$ and $Na^+$ concentrations were determined using a flame photometer (ELEX 6361, Eppendorf AG, Hamburg, Germany). The total hardness (TH) was determined using Eriochrome Black-T ($C_{20}H_{12}N_3O_7SNa$) and ammonium chloride ($NH_4Cl$) indicators against EDTA solution. Titration with silver nitrate ($AgNO_3$) and potassium chromate ($K_2CrO_4$) indicator was used to measure $Cl^-$ concentrations. The titrimetric technique, including a standard solution of sulphuric acid ($H_2SO_4$) and methyl orange indicator, was used to detect $HCO_3^-$, and $CO_3^{2-}$ concentrations. Titration with silver nitrate was used to measure $Cl^-$ concentrations, and the titrimetric technique was used to detect $HCO_3^-$ and $CO_3^{2-}$ concentrations. The concentrations of $SO_4^{2-}$ and $NO_3^-$ were measured using the spectrophotometer instrument based on visible ultraviolet (UV) spectrum (DR/2040- Loveland, CO, USA). Several quality assurance techniques were performed during the examination of the water samples. In view of quality control, validation of the analytical procedures was carried out by proper calibration of instruments and checking their precision and linearity. Charge balance errors (CBE) were calculated after the field measurements were double-checked in the lab, and samples were tested in triplicate, with the average value given. The principle of neutrality states that the summation of cations should be equal to the summation of anions in $meq/L^{-1}$. The error in anion–cation balance is evaluated using Equation (1) [49]. Therefore, the CBE of all the analyzed samples were found within the recommended limit $\pm$ 5%.

$$CBE = \frac{\sum Cations - \sum Anions}{\sum Cations + \sum Anions} \times 100 \qquad (1)$$

Additionally, the analytical procedure's quality assurance was double-checked using Certified Reference Material (CRM) and blank method analysis.

### 2.3. Indexing Method

2.3.1. Water Quality Indices (WQIs)

The appropriateness of groundwater for both drinking and irrigation was assessed using the cited WQIs, which included DWQI, IWQI, TDS, PS, SAR, and RSC values (Table 1). An existing mathematical approach was used to turn the numerical impact of particular values and units of numerous water quality metrics into a single number [50,51], which is used to define the quality of water.

**Table 1.** Arithmetic rating method for calculation of drinking water quality index (DWQI).

| Physicochemical Parameters | Unit | WHO (2017) $S_i$ | Unit Weight $w_i$ | $\frac{C_i}{S_i} \times 100$ | $W_i \times (\frac{C_i}{S_i} \times 100)$ |
|---|---|---|---|---|---|
| pH | - | 8.5 | 0.415 | 86.000 | 35.729 |
| TDS | mg/L | 500 | 0.007 | 105.996 | 0.748 |
| EC | μs/cm | 1500 | 0.002 | 70.600 | 0.166 |
| TH | mg/L | 500 | 0.007 | 60.944 | 0.430 |
| $K^+$ | mg/L | 12 | 0.294 | 65.210 | 19.190 |
| $Na^+$ | mg/L | 200 | 0.017 | 33.388 | 0.589 |
| $Mg^{2-}$ | mg/L | 50 | 0.070 | 34.735 | 2.453 |
| $Ca^{2+}$ | mg/L | 75 | 0.047 | 124.540 | 5.864 |
| $Cl^-$ | mg/L | 250 | 0.014 | 43.076 | 0.608 |
| $SO_4^{2-}$ | mg/L | 250 | 0.014 | 71.666 | 1.012 |
| $HCO_3^{2-}$ | mg/L | 120 | 0.029 | 76.250 | 2.243 |
| $CO_3^-$ | mg/L | 350 | 0.0100 | 3.428 | 0.0345 |
| $NO_3^-$ | mg/L | 50 | 0.070 | 88.775 | 6.270 |
| | | $\sum (S_i)$ | $\sum (w_i) = 1$ | | $\sum_{i=1}^{n} W_i \times (\frac{C_i}{S_i} \times 100)$ |

### 2.3.2. Drinking Water Quality Index (DWQI)

The DWQI values were calculated using the average concentrations of determinants of 30 parameters in Table 1 at each sample location according to Brown et al. [52]. For quality evaluation, the findings of laboratory analyses for all the samples collected were considered. The physicochemical criteria have been weighted according to their relative importance to overall water quality. The WQI depicts the complete water quality of each water component depending on a variety of water quality variables and their application in the environment according to Equation (2):

$$DWQI = \sum_{i=1}^{n} W_i \times \left( \frac{C_i}{S_i} \times 100 \right) \qquad (2)$$

$W_i$ represents each parameter's weight unit, and 13 physicochemical parameters were employed. The computed the concentration ($C_i$) value and standard ($S_i$) for each water parameter according to the following Equation (3):

$$W_i = \frac{wi}{\sum wi} \qquad (3)$$

$w_i$ for each parameter is computed using the recommended standards [53] by Equation (4):

$$w_i = K/S_i \qquad (4)$$

where K denotes the constant of proportionality, which is calculated by using Equation (5):

$$K = \frac{1}{\sum S_i} \qquad (5)$$

To calculate the DWQI, each groundwater parameter ($w_i$) is given a weight, and the relative weight is calculated ($W_i$). Therefore, $W_i$ values were assigned for all physicochemical parameters in Table 1, while $w_i$ was calculated using Equation (4). The computed values of the standards, unit weights ($w_i$), and arithmetic weights ($W_i$) for the water parameters are illustrated in Table 1.

### 2.3.3. Irrigation Water Quality Index (IWQI)

The IWQI was estimated using water quality metrics such as EC, SAR, $Na^+$, $Cl^-$, and $HCO_3^{2-}$ [54–56], according to the following Equation (6):

$$IWQI = \sum_{i=1}^{n} Q_i W_i \qquad (6)$$

Depending on each value of physicochemical parameter, the aggregation weights ($W_i$) and value of water quality parameter ($Q_i$) were calculated using the criteria established by Ayers and Westcot [57] according to Equation (7):

$$Q_i = Q_{max} - \left( \frac{\left[ (X_{ij} - X_{inf}) \times Q_{imap} \right]}{X_{amp}} \right) \qquad (7)$$

where $Q_{max}$ is the greatest $Q_i$ value for each class, $X_{ij}$ is the observed value of each physicochemical parameter, and $X_{inf}$ is the class's lower limit value. $Q_{imap}$ and $X_{amp}$ denote the class amplitude and class amplitude to which the parameter belongs, respectively.

Finally, the $W_i$ values were normalized, and their final total equaled one using the following Equation (8):

$$W_i = \frac{\sum_{j=1}^{k} F_j A_{ij}}{\sum_{j=1}^{k} \sum_{i=1}^{n} F_j A_{ij}} \qquad (8)$$

where Wi and F are the comparative weights of the IWQI physicochemical characteristics, and component i is a constant value; The parameter i that can be described by factor j is denoted by $A_{ij}$. The number of physicochemical parameters used in the IWQI ranges from 1 to n, while the number of factors chosen in the IWQI ranges from 1 to k.

### 2.3.4. Total Dissolves Solids (TDS)

The TDS is an important parameter to express the status of contaminants present in the groundwater, estimated in mg/L [57], according to Equation (9):

$$TDS = (Ca^{2+} + Mg^{2+} + Na^+ + K^+ + Cl^- + SO_4^{2-} + HCO_3^{2-} + CO_3^- + NO_3^-) \quad (9)$$

### 2.3.5. Potential Salinity (PS)

The PS is another irrigation quality index that was estimated in milliequivalents per liter [58] according to Equation (10):

$$PS = Cl^- + (SO_4^{2-}/2) \quad (10)$$

### 2.3.6. Sodium Adsorption Ratio (SAR)

The SAR is an important irrigation quality index, which also evaluates the contents of cations expressed in milliequivalents per liter [59] according to Equation (11):

$$SAR = \left( \frac{Na^+}{\sqrt{\left(Ca^{2+} + Mg^{2+}\right)/2}} \right) \times 100 \quad (11)$$

### 2.3.7. Residual Sodium Carbonate (RSC)

The RSC is another index to assess the suitability of water for irrigation, which is expressed as Equation (12) [60,61].

$$RSC = (HCO_3^{2-} + CO_3^-) - (Ca^{2+} + Mg^{2+}) \quad (12)$$

### 2.4. Partial Least-Square Regression (PLSR) and Multiple Linear Regression (MLR)

In this work, PLSR models were used to evaluate the WQIS, DWQI, IWQI, TDS, SAR, PS, and RSC. PLSR models of six WQIs were constructed using version 10.2 of the unscramble X program (CAMO Software AS, Oslo, Norway). The PLSR model used physicochemical parameters in Table 1 as the input parameter (independent parameters) to predict the DWQI as output parameters (dependent v parameters). The PLSR model also used chemical parameters in Table 1 as the input parameter (independent parameters) to predict the IWQI, TDS, SAR, PS, and RSC as output parameters (dependent parameters). The input variables were linked to the output variables using PLSR and leave-one-out cross-validation (LOOCV). In PLSR analysis, selecting the correct number of latent variables (LVs) to represent the calibration data without overfitting or underfitting is critical. Random 10-fold cross-validation was carried out on the datasets to improve the robustness of the results.

Four criteria were used to evaluate the PLSR's performance in predicting the six WQIs for calibration (Cal.) and validation (Val.) models.

(1)  $R^2$ coefficient;

$$R^2 = 1 - \frac{\sum_{i=1}^{n} (WQIso_i - WQIs_{fi})^2}{\sum_{i=1}^{n} (PIso_i)^2} \quad (13)$$

(2)  root mean square error (RMSE);

$$\text{RMSE} = \sqrt{\frac{\sum_{i=1}^{n}(\text{WQIso}_i - \text{WQIs}_{fi})^2}{n}} \tag{14}$$

(3)   mean absolute deviation (MAD);

$$\text{MAD} = \frac{\sum_{i=1}^{n}|\text{WQIso}_i - \text{WQIs}_{fi}|}{n} \tag{15}$$

(4)   Accuracy (ACC) of the models

$$\text{Acc} = 1 - \text{abs}\left(\text{mean}\,\frac{\text{WQIs}_{fi} - \text{WQIso}_i}{\text{WQIso}_i}\right) \tag{16}$$

e measured value is $\text{WQIso}_i$, the number of data points is n, and the predicted value is $\text{WQIs}_{fi}$. The best models were chosen for their low RMSE and MAD, as well as their high $R^2$ and Acc.

### 2.5. Data Analysis and Graphical Approach

SPSS version 22 was used to construct a statistical analysis (range, mean, standard deviation) of the physical and chemical characteristics (SPSS Inc., Chicago, IL, USA). Different models for hydrochemical facies evolution, such as the Piper, Chadah, Gibbs, and hydrochemical facies evolution diagrams [62–65], have been proposed utilizing Geochemist's Software package 12.0 to determine water types, geochemistry mechanisms, and major water chemical control factors. To generate zoning maps for water quality indicators assessed in the present study, GIS version 10.0 was utilized. The DWQI and IWQI maps were constructed by combining datasets for physicochemical metrics because of geo-statistical data analysis, which included the use of inverse distance weighted (IDW) kriging. This method is a component enhancement for the spatial analytical technique in GIS. In addition, the CA and PCA were created by dividing a set of variables by their maximum values using PAST software (V. 4.0) in order to understand the relationships and variance between the physicochemical determinants.

## 3. Results and Discussion

### 3.1. Physicochemical Parameters

Physical and chemical metrics are useful benchmarks to understand the status of water geochemistry and associated regulatory processes, and therefore play a crucial role in the evolution of water quality. Table 2 contains statistical descriptions of the physical and chemical characteristics in the analyzed groundwater points (min., max., mean, and standard deviation).

**Table 2.** Statistical description of several physical and chemical parameters in the collected groundwater wells.

| | T °C | pH | EC | TDS | TH | K⁺ | Na⁺ | Mg²⁺ | Ca²⁺ | Cl⁻ | SO₄²⁻ | HCO₃²⁻ | CO₃⁻ | NO₃⁻ |
|---|---|---|---|---|---|---|---|---|---|---|---|---|---|---|
| | $T\,°C$ | $pH$ | $EC$ | $TDS$ | $TH$ | $K^+$ | $Na^+$ | $Mg^{2+}$ | $Ca^{2+}$ | $Cl^-$ | $SO_4^{2-}$ | $HCO_3^{2-}$ | $CO_3^-$ | $NO_3^-$ |
| Min | 30.00 | 6.99 | 553.00 | 226.90 | 44.10 | 0.99 | 43.64 | 4.11 | 10.91 | 70.53 | 30.00 | 12.20 | 0.00 | 0.01 |
| Max | 32.00 | 8.39 | 25,000.00 | 18,518.30 | 6025.50 | 79.03 | 4602.75 | 575.27 | 1995.80 | 7271.03 | 5180.27 | 274.50 | 24.00 | 475.44 |
| Mean | 30.60 | 7.74 | 4217.20 | 2572.30 | 1188.90 | 13.87 | 441.93 | 90.56 | 327.25 | 926.22 | 692.35 | 146.18 | 7.01 | 57.27 |
| SD | 0.75 | 0.33 | 4595.60 | 3247.10 | 1209.10 | 13.35 | 729.02 | 108.44 | 333.83 | 1450.50 | 788.69 | 51.56 | 8.19 | 84.55 |

Makkah Al-Mukarramah Province, KSA (*n* = 59)

All physical and chemical parameters are stated in mg/L excluding temperature (T °C), pH and EC (μs/cm).

The data analysis of physicochemical characteristics for groundwater samples obtained showed that pH values varied between 6.99 and 8.39 by a mean of 7.74, which indicated that the groundwater was slightly alkaline. Groundwater temperatures ranged from 30.00 to 32.00 °C, depending on the depths to water surface in wells (Figure 3). The EC readings ranged from 553 to 25,000 μs/cm, with an average value of 4217.20 μs/cm. TDS levels varied from 226.90 to 18,518.30 mg/L, with a mean value of 2572.3 mg/L, which reflected brack-

ish groundwater type. The ionic content of $K^+$, $Na^{2+}$, $Mg^{+2}$, $Ca^{+2}$, $Cl^-$, $SO_4^{2-}$, $HCO_3^{2-}$, $CO_3^-$ and $NO_3^-$ displayed mean values of 13.87, 441.93, 90.56, 327.25, 926.22, 692.35, 146.18, 7.01 and 57.27 mg/L, respectively (Table 2). Therefore, the average values of ions presented sequences of $Na^{2+} > Ca^{+2} > Mg^{+2} > K^+$, and $Cl^- > SO_4^{2-} > HCO_3^{2-} > NO_3^- > CO_3^-$, respectively. These values indicated that $Na^{2+}$ was the dominant cation and $Cl^-$ was the dominant anion in the collected water samples.

### 3.2. Geochemical Facies and Controlling Mechanisms

Hydrochemical data were evaluated using imitative techniques to better understand the numerous geochemical processes that regulate groundwater quality. For identifying the geochemical facies and types of groundwater in the study area, Piper's trilinear diagram was used to determine the prevailing cations and anions in meq/L of the collected samples (Figure 4a). The chemical properties of the examined groundwater samples revealed that the hydrochemical facies were $Ca-HCO_3$, Na-Cl, mixed $Ca-Mg-Cl-SO_4$, and $Na-Ca-HCO_3$. Chadah's arrangement is also used to determine the hydrochemical mechanisms and groundwater types (Figure 4b). The groundwater samples were scattered in fields 2 and 3, which demonstrated a reverse ion exchange process as a result of cation exchange process in the groundwater system and mixing with saline water, especially downstream of Wadi Fatimah. As a result, $Ca^{2+}$ in the groundwater was replaced by $Na^+$ in the aquifer, which reflected a decrease in $Ca^{2+}$ concentration and increase in $Na^+$ concentration, indicating that groundwater in the study area was affected by the cation exchange process.

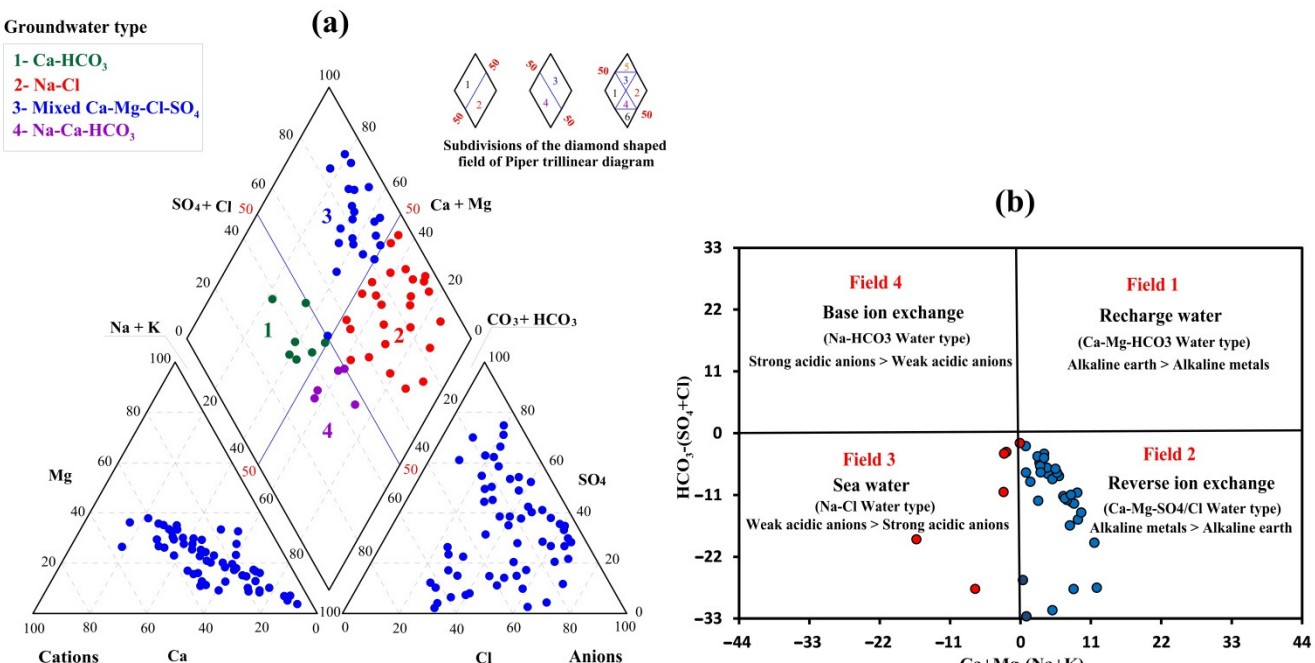

**Figure 4.** Geochemical facies and water type: (**a**) Piper diagram and (**b**) Chadha diagram.

The effects of the weathering process and aquifer matrix considerably alter the chemical composition of groundwater. The Gibbs diagram is frequently applied to create the link between the water component and the aquifer matrix [64] (Gibbs, 1970). The key regulating processes that determine groundwater geochemistry are identified by Gibb's diagram through displaying TDS vs. (Na + K)/(Na + K + Ca) and Cl/(Cl + $HCO_3$). As shown in a plotting of chemical data on the Gibbs diagram, groundwater samples were spread in the evaporation, weathering, and rock–water interaction fields (Figure 5a), which are significant processes regulating groundwater chemistry and quality. HFE plot findings demonstrated dissolving of evaporite from salt marches in aquifer materials, which are high in sulphate and chloride (Figure 5b) content. The majority of samples representing mixed

water (Ca-Mg-HCO$_3$ and Na-Cl) revealed high calcium and sodium content, as a result of clays in the Quaternary aquifer, and the volcaniclastic sequence of the Fatimah Basin was found to be rich in Na$^+$ [66] and the groundwater in Ca$^{2+}$. Obviously, most groundwater samples scattered in the intrusion area indicated that intrusion is a significant process in the formation of dissolved solutes for groundwater samples. Moreover, evaporation is the most significant mechanism for groundwater and soil salinization in areas with shallow groundwater depth as well as high evaporation rate [67]. Brines from evaporate minerals dissolve in the recharge zone in closed basins, enhancing the salinity of the groundwater over time. In closed basins, salinity levels increase from the inflow area to the outflow area as a result of over-pumping and a negative water balance, converting the hydrochemical facies from Ca-HCO$_3$ to Na-Cl [68]. The chemical composition changes are mostly induced via groundwater flow direction and reactions [69].

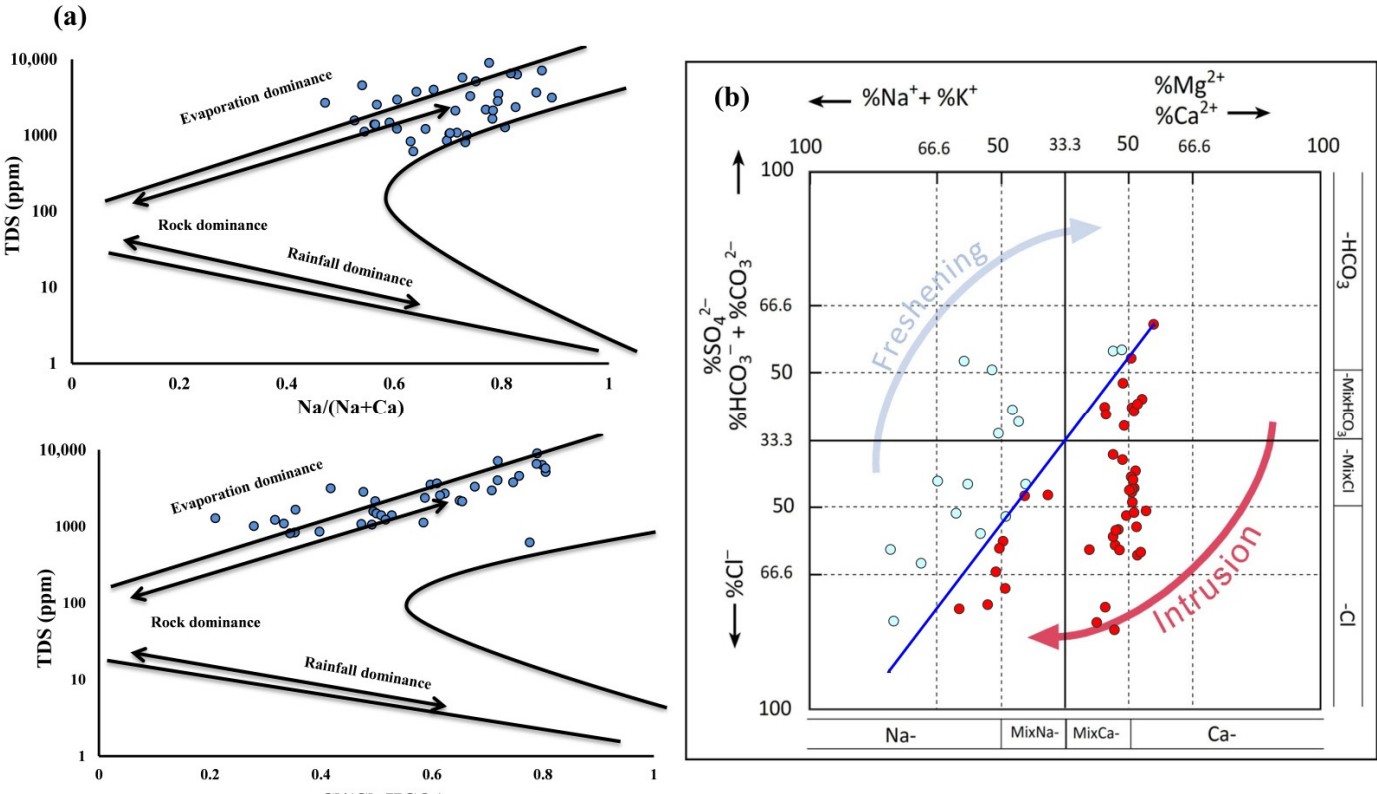

**Figure 5.** Geochemical controlling mechanisms: (**a**) Gibbs diagram and (**b**) Hydrochemical facies evolution diagram (HFE).

### 3.3. Water Quality Indices

Table 3 showed the statistical analysis and categorization of several WQIs in this study, which included DWQI, IWQI, TDS, PS, SAR, and RSC. In addition, GIS-Zoning maps for each index were used to display and examine the quality of water in the investigated area for potable and agricultural purposes (Figures 6–8).

**Table 3.** Statistical analyses and classification of the different water quality indices (WQIs).

| Water Quality Indices (WQIs) | Sample Range | | | | Range | Water Category | Number of Samples (%) |
|---|---|---|---|---|---|---|---|
| | Min. | Max. | Mean | SD | | | |
| Drinking water quality (DWQI) | 23.29 | 545.53 | 118.68 | 88.28 | 0–25 | Excellent | 1 (2%) |
| | | | | | 26–50 | Good | 2 (3%) |
| | | | | | 51–75 | Poor | 10 (17%) |
| | | | | | 76–100 | Very poor | 19 (32%) |
| | | | | | >100 | Unsuitable | 27 (46%) |
| Irrigation water quality index (IWQI) | 19.42 | 95.93 | 64.07 | 20.42 | 85–100 | No restriction | 15 (25.5%) |
| | | | | | 70–85 | Low restriction | 12 (20.5%) |
| | | | | | 55–70 | Moderate restriction | 5 (8.5%) |
| | | | | | 40–55 | High restriction | 22 (37%) |
| | | | | | 0–40 | Serve restriction | 5 (8.5%) |
| Total dissolved solids (TDS) | 226.90 | 18,518.32 | 2572.30 | 3247.14 | <700 | No restriction | 12 (20%) |
| | | | | | 700–3000 | Slight to moderate restriction | 33 (56%) |
| | | | | | >3000 | Serve restriction | 14 (24%) |
| Potential salinity (PS) | 3.03 | 258.96 | 33.32 | 48.17 | <3 | Excellent to good | 0 (0.0%) |
| | | | | | 3 to 5 | Good to Injurious | 8 (14%) |
| | | | | | >5 | Injurious to Unsatisfactory | 51 (86%) |
| Sodium adsorption ratio (SAR) | 1.109 | 31.00 | 4.84 | 4.86 | 2–10 | Excellent | 54 (92%) |
| | | | | | 10–18 | Good | 3 (5%) |
| | | | | | 18–26 | Doubtful or Fairly poor | 1 (1.5%) |
| | | | | | >26 | Unsuitable | 1 (1.5%) |
| Residual Sodium Carbonate (RSC) | −118.91 | −0.40 | −21.14 | 24.33 | <1.25 | Safe | 59 (100%) |
| | | | | | 1.25–2.5 | Marginal | 0 (0.0%) |
| | | | | | >2.5 | Unsuitable | 0 (0.0%) |

All WQIs are estimated in meq/L except DWQI, IWQI and TDS in mg/L. Min.: Minimum, Max.: Maximum, SD: Standard deviation.

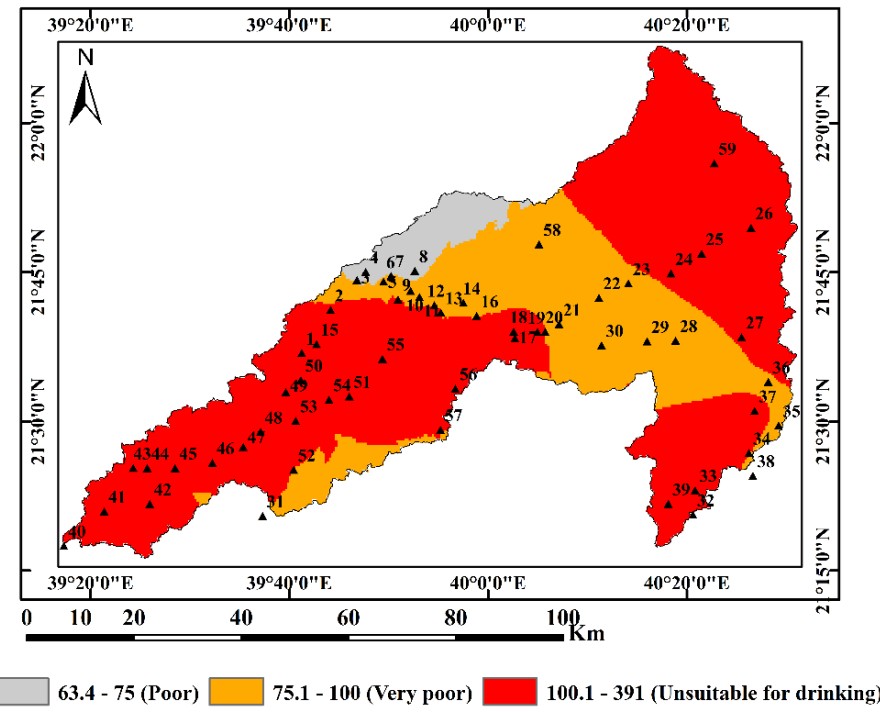

**Figure 6.** Spatial distribution map of DWQI in the study area.

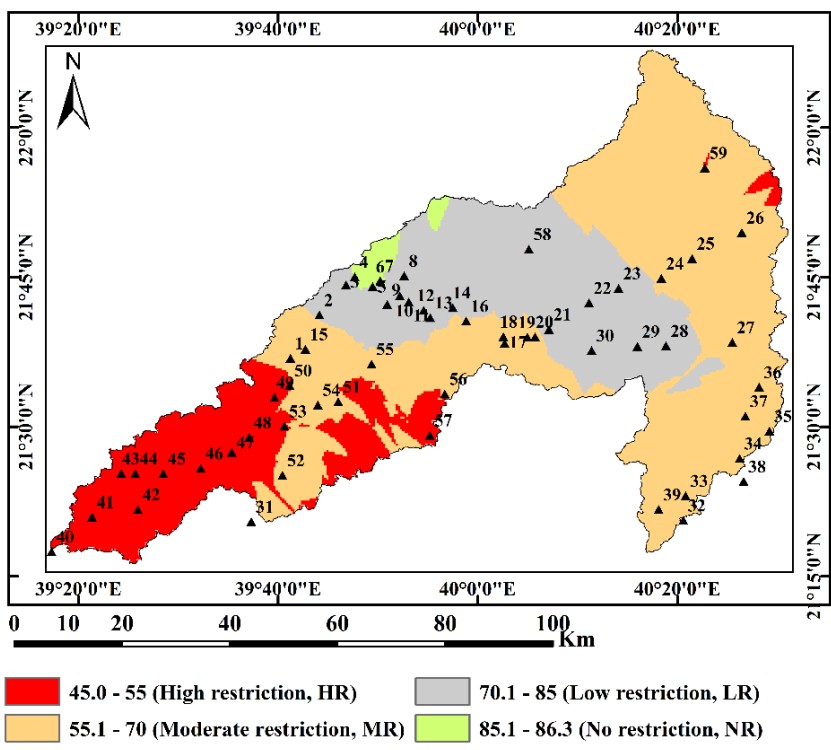

**Figure 7.** Spatial distribution map of IWQI in the study area.

### 3.3.1. Drinking Water Quality Index (DWQI)

The DWQI model was applied to determine groundwater quality, which is classified based on the purity level of routinely detected water quality parameters according to Equation (1). The DWQI was developed to measure the acceptability of groundwater for drinking. The computed value of DWQI in the obtained groundwater samples is shown in Table 4, ranging from 23.29 to 545.53, with an average of around 118.68. According to DWQI categorization (Table 3), approximately 2% of groundwater was categorized as excellent, 3% as good, 17% as poor, 32% as extremely poor, and 46% as unfit for drinking. The DWQI distribution map (Figure 6) indicates that most of the groundwater samples cannot be utilized for safe drinking due to evaporation, rock–water interaction, and reverse ion exchange process in the north-eastern parts as well as saltwater intrusion downstream of Wadi Fatimah toward the Red Sea.

**Table 4.** Results of calibration ($R^2_{cal}$, $RMSE_C$, $MAD_c$, and $Acc_c$), and ten-fold cross-validation ($R^2_{val}$, $RMSE_v$, $MAD_v$, and $Acc_v$): PLSR models of the relationships between several physicochemical parameters and drinking water quality index (DWQI), irrigation water quality index (IWQI), total dissolved solids, potential salinity (PS), sodium absorption ratio (SAR), and residual sodium carbonate (RSC). ***: $p < 0.001$.

| | Water Quality Indices | LVs | Calibration | | | | Validation | | | |
|---|---|---|---|---|---|---|---|---|---|---|
| | | | $R^2_{cal}$ | $RMSE_c$ | $MAD_c$ | $ACC_c$ | $R^2_{val}$ | $RMSE_v$ | $MAD_v$ | $ACC_v$ |
| PLSR | DWQI | 9 | 0.992 *** | 7.356 | 6.323 | 0.991 | 0.987 *** | 10.030 | 7.686 | 0.989 |
| | IWQI | 4 | 0.905 *** | 10.516 | 5.781 | 0.999 | 0.848 *** | 13.680 | 6.331 | 0.984 |
| | TDS | 2 | 0.999 *** | 58.920 | 21.147 | 0.981 | 0.999 *** | 71.985 | 26.538 | 0.980 |
| | PS | 1 | 0.989 *** | 0.982 | 0.279 | 0.998 | 0.999 *** | 1.494 | 0.273 | 0.985 |
| | SAR | 2 | 0.919 *** | 1.370 | 0.811 | 0.824 | 0.861 *** | 1.838 | 0.814 | 0.817 |
| | RSC | 2 | 0.999 *** | 0.762 | 0.427 | 0.962 | 0.998 *** | 0.874 | 0.448 | 0.924 |

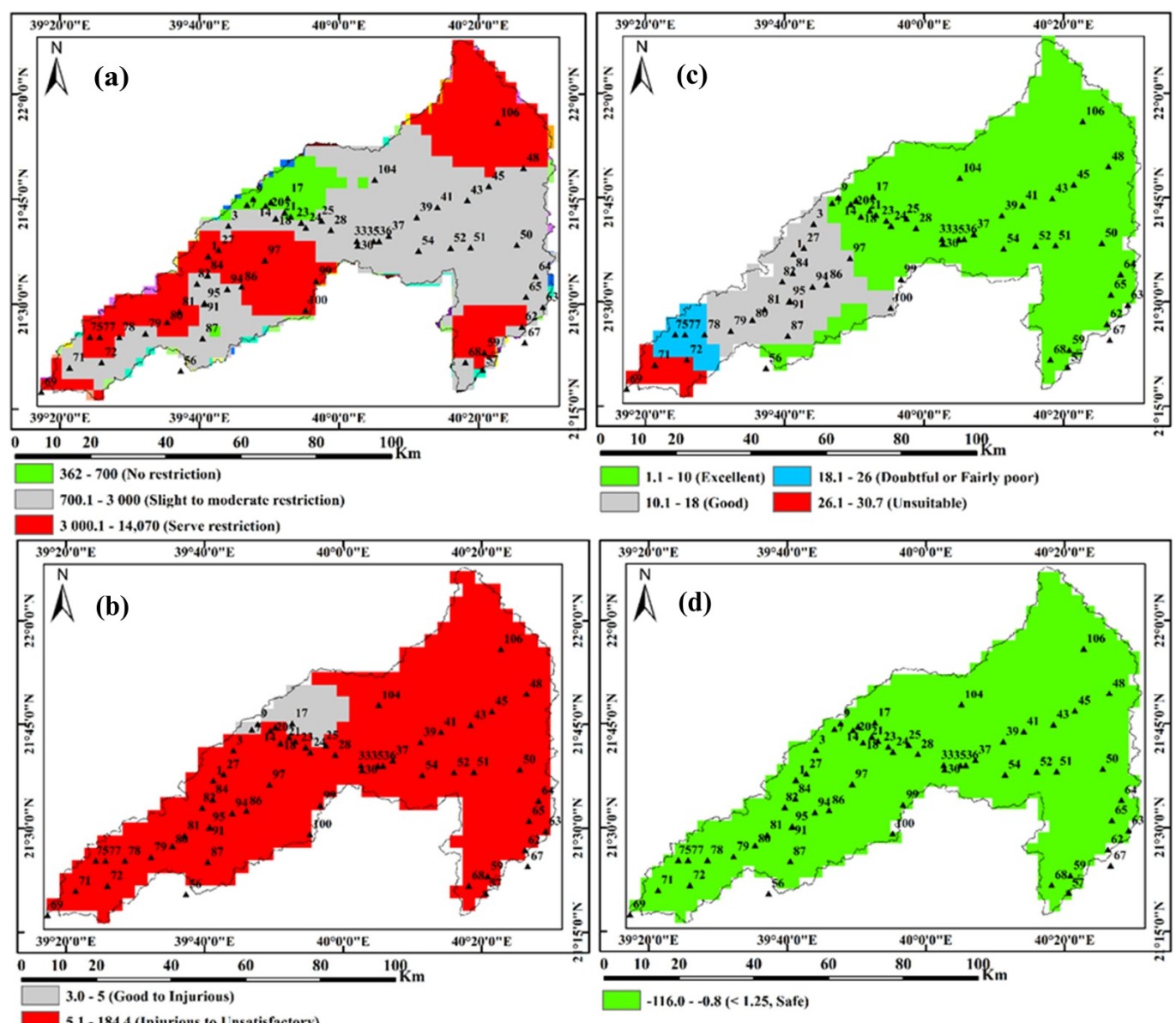

**Figure 8.** Spatial distribution maps of irrigation water quality indices: (**a**) total dissolved solids (TDS), (**b**) potential salinity (PS), (**c**) sodium adsorption ratio (SAR), and (**d**) residual sodium carbonate (RSC).

### 3.3.2. Irrigation Water Quality Index (IWQI)

The IWQI was recognized as one of the most essential methods for urban planners to analyze irrigated water quality since it provided a clear categorization of water quality based on its effect on soil and plants [70]. According to the results of IWQI classification (Table 3 and Figure 7), about 8.5% of the examined groundwater samples fall in the severe restriction range, which may be utilized to irrigate high salt sensitivity crops. While 37% of the wells studied fall in the high restriction category, which can be utilized to irrigate moderate to high salinity tolerance crops in loose soil with no compacted layers and a specific frequency of irrigation (EC > 2000 S/cm and SAR > 7), approximately 8.5% of the tested wells fall in moderate limitation, which may be utilized to irrigate moderate salinity tolerance crops and are suggested for medium to high permeable soils with respect to leaching processes. Furthermore, 20.5% of the samples fall in the low restriction group, which suggests preventing salinity tolerance crops with respect to irrigated soil characteristics, permeability, and soil sodicity hazards. Lastly, 25.5% of wells were detected with no limitation range and may be used for most soils with potentially low harmful effects

of salinity and sodicity on most crops [57]. Figure 7 showed the spatial distribution of different water quality types for irrigation with respect to IWQI, where the quality of the groundwater for agriculture decreased greatly from the northeast to southwest toward the Red Sea with the same groundwater flow direction.

### 3.3.3. Total Dissolved Solids (TDS)

TDS is commonly used to determine the salinity of groundwater wells, which in this case ranged from 226.90 to 18,518.32 with an average of 2572.30. Table 3 and Figure 8a show the categorization of collected groundwater wells according to salinity levels. TDS values indicated that about 20% of the groundwater wells have salinity levels less than 700 mg/L, which ensures no restriction for irrigation; approximately 56% have salinity levels ranging from 700 to 3000 mg/L, which mandates slight to moderate restriction for agriculture; and the remaining 24% have a salinity level of more than 3000 mg/L, making it unsuitable and highly restrictive for irrigation [71–73], as presented in Table 3. Figure 8a showed the spatial distribution of groundwater quality for agriculture with respect to salinity hazard in the investigated area. As a result, the use of groundwater for irrigation becomes more restricted downstream of Wadi Fatimah.

### 3.3.4. Potential Salinity (PS)

The PS, computed as the totality of $Cl^-$ and half of $SO_4^{2-}$, is also an essential metric for measuring the appropriateness of groundwater for irrigation. The values of PS varied from 3.03 to 258.96 with an average of 33.32 (Table 3). Approximately 14% of the total samples were categorized as good to injurious, while 86% of total samples were categorized as injurious to unsatisfactory for agriculture. Figure 8b showed spatial distribution of groundwater quality for agriculture with respect to PS, where high PS was observed in most parts of the investigated area (Figure 8b).

### 3.3.5. Sodium Absorption Ratio (SAR)

SAR is a ratio of the primary alkaline and earth alkaline cations available in water to crops. It is a useful measurement for evaluating the acceptability of irrigated water depending on sodium risk [74,75], and is more strongly connected to the exchangeable sodium percentages of the soil [76]. The use of high-sodium water for agriculture may enhance the interchange of sodium levels in the soil, reducing soil permeability and soil structure [77]. Soil treatment may be required in agriculture, where the water has a high SAR value to minimize long-term soil deterioration because the $Ca^{2+}$ and $Mg^{2+}$ in the soil may be displaced by sodium in the water. It may also result in decreased soil penetration and permeability to water, which may be hazardous for crop productivity. The water quality categorization for irrigation according to SAR (Table 3) indicated that the SAR value ranged from 1.109 to 31.0 with an average of 4.84. Table 3 showed that around 92% of the wells were found in the range of excellent category and about 5% in the range of good category, thereby suitable for irrigation purposes with no alkali hazard to the crops. The rest of the samples, about 3%, ranged from fairly poor to unsuitable for irrigation. The SAR spatial distribution map (Figure 8c) shows that groundwater quality for irrigation decreases gradually from the northeast to southwest direction toward the Red Sea, compatible with the direction of groundwater flow in the area.

### 3.3.6. Residual Sodium Carbonate (RSC)

Alkalinity concentration of water is a significant factor in evaluating its appropriateness for irrigation [78]. The term 'Residual Sodium Carbonate' (RSC) is used when alkalinity contents exceed alkaline earth metals ($Ca^{2+}$ and $Mg^{2+}$) and shows the harmful influence of alkalinity on irrigation water quality [79,80]. The RSC of groundwater wells varied from −118.91 to −0.40 with an average of −21.14 (Table 3). Results showed that all samples had RSC values of less than 1.25, indicating minimal alkalinity hazard and

the ability to be utilized safely for agriculture with no development of alkalinity hazard (Figure 8d).

### 3.4. Multivariate Statistical Analysis

The CA and PCA were used in multivariate statistical analysis to detect the sources accountable for changes in the quality of water resource by combining primary variables into a new set of variables. CA results for major physical and chemical parameters indicated three kinds of clustering (Figure 9a), with EC and TDS in the same cluster (Cluster 1); $Na^+$, $SO_4^{2-}$, and $Cl^-$ in another cluster (Cluster II); and pH, $K^+$, $Ca^{2+}$, $Mg^{2+}$, $HCO_3^{2-}$, $CO_3^{2-}$, and $NO_3^-$ in a different cluster (Cluster III). According to the CA of major physicochemical parameters, groundwater in the study area was categorized into $Na^+$, $Ca^{2+}$, $Mg^{2+}$, and $K^+$ as dominant cations, and $Cl^-$, $SO_4^{2-}$, and $HCO_3^{2-}$ as dominant anions respectively (Figure 9a). According to the CA results, the high content of $Na^+$ and $Ca^{2+}$ suggested rock–water interaction that revealed the release of $Ca^{2+}$ by the weathering of silicate minerals, while the high concentration of $Cl^-$ and $SO_4^{2-}$ revealed clay interaction with aquifer matrix and saltwater intrusion. The distribution of physicochemical parameters in the groundwater resources revealed the second and final stages of water evolution, which reflects deterioration in groundwater quality in the investigated area as a result of evaporation process, saltwater intrusion, weathering process, and rock–water interaction. These results are in agreement with the water facies presented by Piper plotting due to the effects of evaporation, weathering, and rock–water interactions stated in the Gibbs and Chadha diagrams.

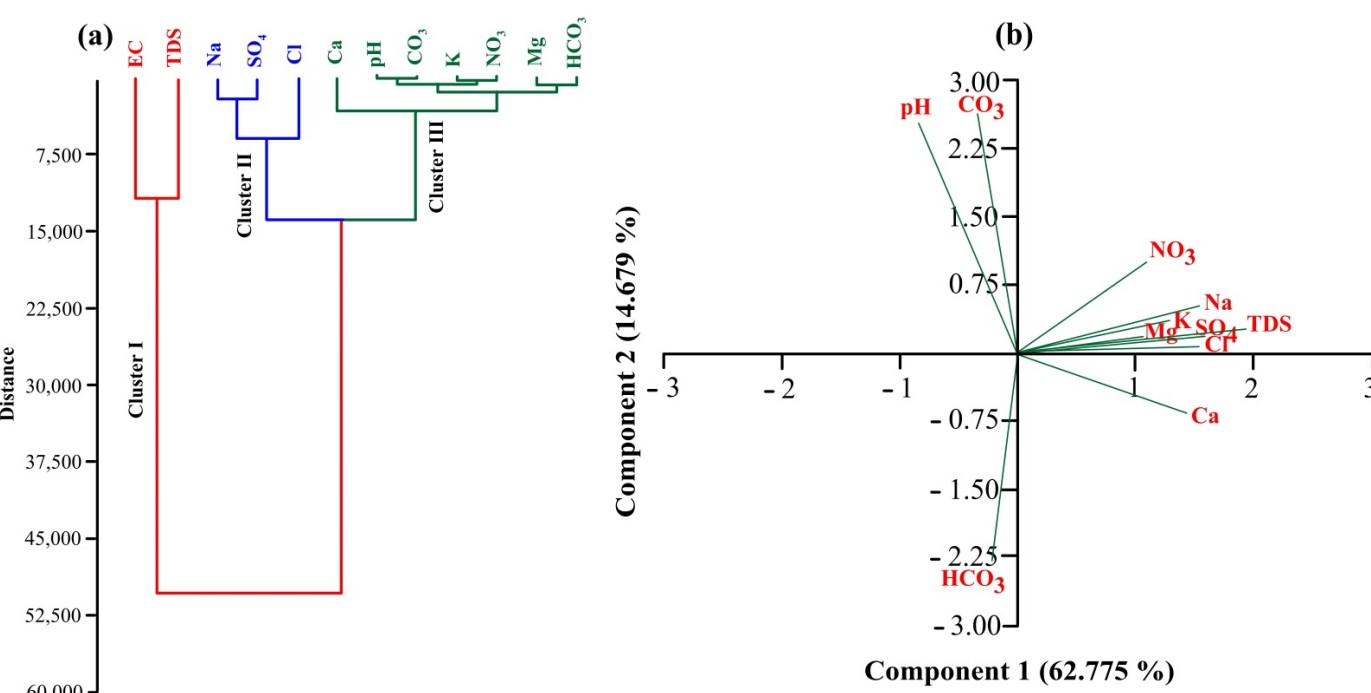

**Figure 9.** Multivariate statistical analysis for physicochemical parameters of groundwater samples in Wadi Fatimah basin: (**a**) Cluster analysis and (**b**) Principal component analysis.

According to the PCA results for physicochemical features of major ions in the groundwater samples obtained, large positive loadings of $Na^+$, $Ca^{2+}$, $Mg^{2+}$, $K^+$, $Cl^-$, $SO_4^{2-}$, and $NO_3^-$ prevailed over PC1 in explaining 62.775% of total variance while PC1 explained 14.679% of the total variance that prevailed by loading of pH, $HCO_3^{2-}$, and $CO_3^{2-}$ (Figure 9b). The majority of cations and anions were clustered together in positive-loading combinations except for total alkalinity ($HCO_3^{2-}$ and $CO_3^{2-}$), which indicated that the variables have a substantial association. According to PCA results, the presence of nine principal components demonstrated the influence of significant ions on groundwater

quality in the studied region. Therefore, PC1 presented maximum loading of $Na^+$, $Ca^{2+}$, $Cl^-$, and $SO_4^{2-}$, while PC2 presented maximum loading of pH, $HCO_3^{2-}$, and $CO_3^{2-}$. These findings could be attributable to evaporation, weathering, saltwater intrusion, and rock–water interaction. Most groundwater samples in the investigated area were highly contaminated with $NO_3^-$, as shown by the high positive loading of $NO_3^-$ in PC1, which revealed agriculture runoff and the effects of the study region being surrounded by urban sewage flowing through the estuary. Therefore, combining physicochemical characteristics in the PCA for groundwater quality assessment is a helpful and adaptable method with remarkable potential and unique insights.

### 3.5. Using Partial Least Square Regression to Predict WQIs for Drinking and Irrigation

PLSR (Partial Least Square Regression) is a reliable method for modeling complex non-linear interactions, especially when the relationships between variables are ambiguous. In this investigation, PLSR models were utilized to forecast DWQI based on physicochemical parameters and five irrigation indices (IWQI, TDS, SAR, PS and RSC) based on chemical parameters as illustrated in Table 2. The classical mathematical methods mentioned in this study can be used to produce approximate predictions of the DWQI and five IWQIs of the water samples [52,57–60,81]. In this work, PLSR was investigated as an alternative method for predicting WQIs, considering that it is quick, uncomplicated, and does not require many steps to calculate especially the DWQI and IWQI. In addition, PLSR can be used to select the most effective parameters for calculating DWQI and IWQI. This, in turn, leads to reducing the number of elements that were used in the chemical analysis to calculate WQIs and, finally, decreasing the overall cost. The number and influence of input factors have a big impact on the exact forecast, but all data must be available and cost-effective. Based on many response variables, the PLSR predicts a single model [45,82,83].

Figures 10 and 11 illustrate the relationships of DWQI and five IWQIs between observed and predicted values in a 1:1 scatter plot using PLSR for the Cal. and Val. models. PLSR presented accurate prediction models for WQIs in Cal. and Val. For example, the PLSR models of all IWQs had determination coefficient values of $R^2$ ranging from 0.905 to 0.999 in the Cal., and ranging between 0.848 and 0.999 in the Val. datasets (Table 4), and had model accuracy ranging from 0.824 to 0.999 in the Cal., and ranging from 0.817 to 0.989 in the Val. dataset. The RMSE values for DWQI, IWQI, TDS, SAR, PS, and RSC were 7.356, 10.516, 58.920, 0.982, 1.370, and 0.762 in the Cal. dataset, respectively, and were 10.030, 13.680, 71.985, 1.494, 1.838, and 0.87 in the Val. dataset, respectively. The PCs were designated to support the calibration data without over-fitting for the PLSR models of six WQIS, and it ranged from 1 to 9 (Table 4). Similar to this study's prediction of WQIs, Gad et al. [45] discovered that PLSR could be utilized to estimate the DWQI and three surface water pollution indices in the Northern Nile Delta. Elsayed et al. [84] found that the multivariate method of Principal Component Regression (PCR) and machine learning of Support Vector Machine Regression SVMR revealed accurate estimation and produced robust models for forecasting the WQIs in both (Cal.) and (Val.), and they had $R^2$ values varying from 0.48 to 0.99 in the Northern Nile Delta, Egypt. Abowaly et al. [44] recently discovered that the PLSR and multiple linear regression (MLR) models performed the best in predicting the PLI of the soil based on data for the four examined elements, with $R^2$ 0.92–0.94 across the three layers. In general, the PLSR models produced strong and reliable estimates of different indices, with the highest $R^2$ and highest slope values near 1.00 as well as the lowest RMSE values in both models.

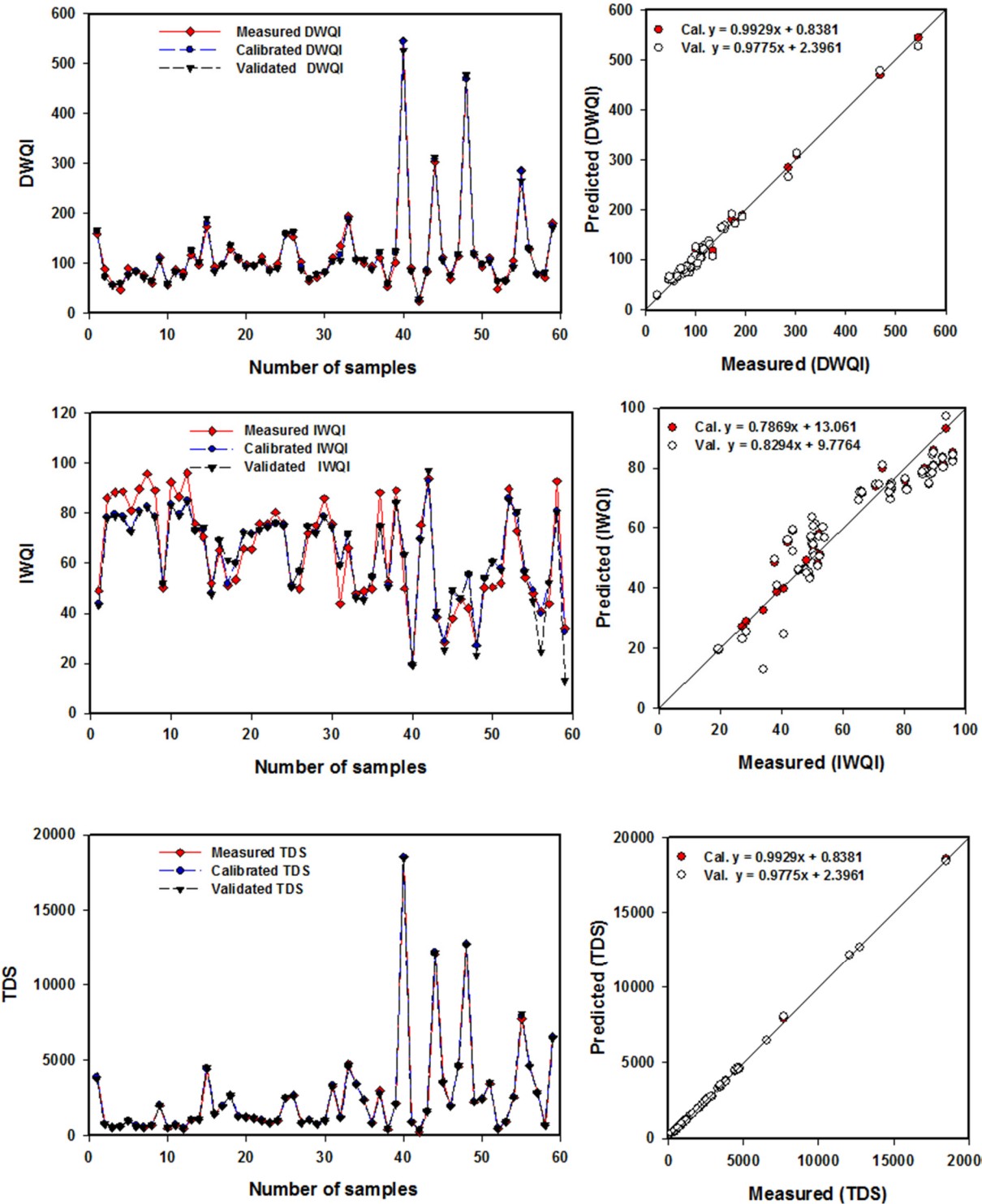

**Figure 10.** Comparison between measuring datasets, calibrating datasets, and validating datasets for drinking water quality index (DWQI), irrigation water quality index (IWQI), and total dissolved solids (TDS) using PLSR models based on several physicochemical parameters.

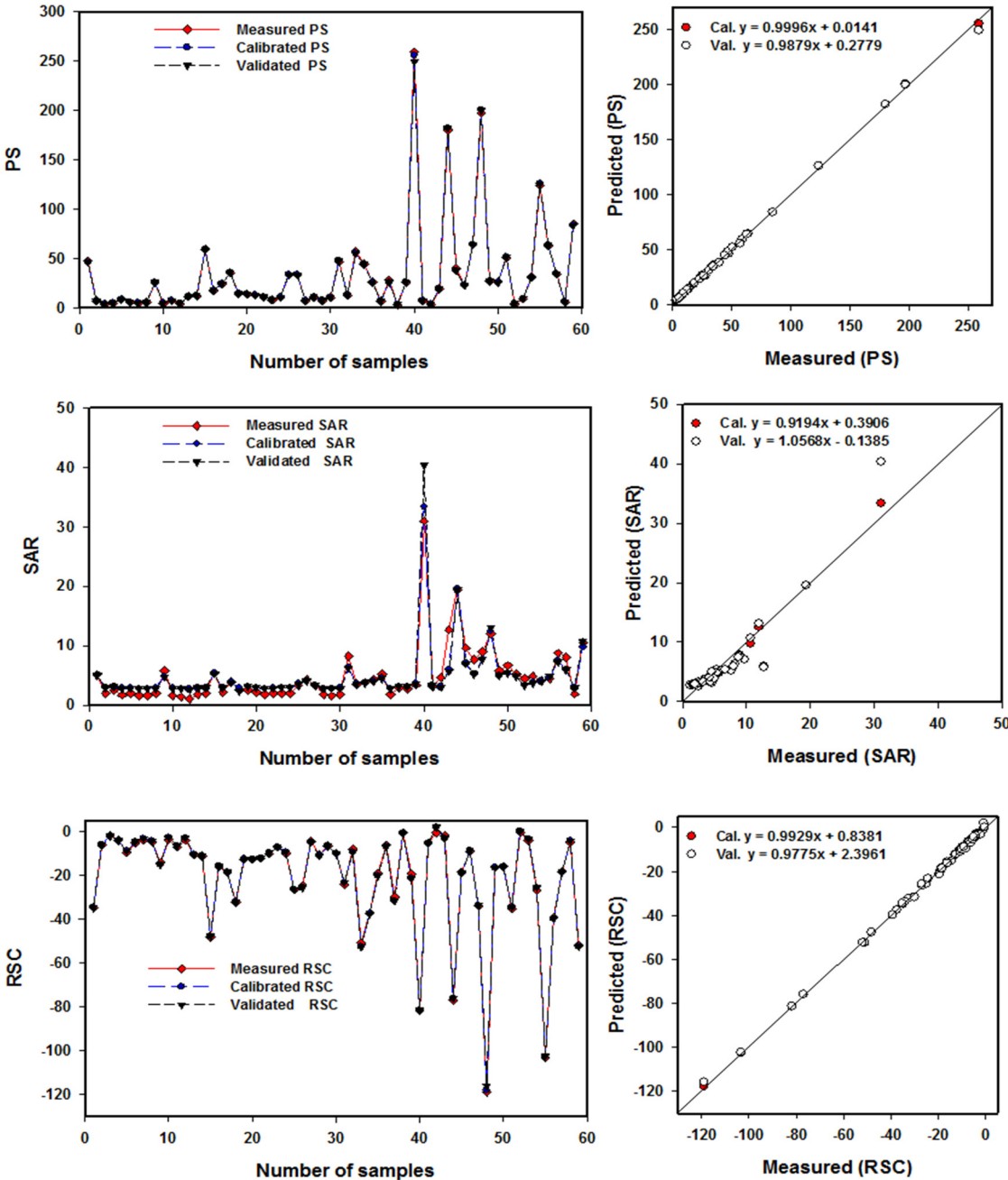

**Figure 11.** Comparison between measuring datasets, calibrating datasets, and validating datasets for potential salinity (PS), sodium absorption ratio (SAR) and residual sodium carbonate (RSC) using PLSR models based on several physicochemical parameters.

## 4. Conclusions

This study examined the suitability of groundwater in Wadi Fatimah, Saudi Arabia, for drinking and irrigation purposes. Physicochemical characteristics, water quality indices, multivariate modeling as well as GIS techniques were coupled to investigate hydrogeochemical characteristics of the Quaternary groundwater aquifer and corresponding geochemical facies and controlling factors. The analytical results of major ions exhibited the trends of $Na^{2+} > Ca^{+2} > Mg^{+2} > K^+$, and $Cl^- > SO_4^{2-} > HCO_3^{2-} > NO_3^- > CO_3^-$, respectively. These trends revealed that the hydrochemical facies were $Ca\text{-}HCO_3$, $Na\text{-}Cl$, mixed $Ca\text{-}Mg\text{-}Cl\text{-}SO_4$, and $Na\text{-}Ca\text{-}HCO_3$. According to the geochemical properties of groundwater, evaporation and saltwater intrusion were the predominant factors controlling the quality of groundwater in the region. DWQI values indicated that the majority

of inventoried wells, about 95%, varied between poor to unsuitable class for drinking, requiring proper treatment before use and a water management strategy. IWQI values indicated that about 45.5 % of the samples varied between high to severe restriction class for irrigation use, which can be utilized for the agriculture of high saline sensitivity crops, while 54.5% of samples varied from moderate to no restriction for irrigation. Agriculture indices like total dissolved solids (TDS), potential salinity (PS), sodium absorption ratio (SAR), and residual sodium carbonate (RSC) showed the mean values of 2572.30, 33.32, 4.84, and −21.14, respectively. However, the quality of the groundwater in the study area improves with increased rainfall and thus recharging the Quaternary aquifer.

By calibrating and validating the data sets, the PLSR models were implemented well in estimating the DWQI, IWQI, TDS, SAR, PS, and RSC, with the highest $R^2$, lowest RMSE and MAD values, and highest slope values. For the PLSR models of six WQIs, there were no apparent overfitting or underfitting between measuring, calibrating, and validating datasets. So, a comprehensive picture of water quality and governing mechanisms can be obtained by integrating physicochemical data, WQIs, multivariate modeling, and GIS tools. Therefore, the use of different techniques and indicators that cross-validate was recommended for assessing water quality for general and specific utilization.

**Author Contributions:** Conceptualization, M.E.O., M.G., S.E., M.M. and A.A.; fieldwork, M.E.O., A.A. and M.M.; methodology, M.G., S.E., M.M. and M.E.O.; software, S.E., M.M. and M.G.; validation, S.E., M.M. and M.E.O.; formal analysis, S.E. and M.E.O.; investigation, M.E.O., A.A. and M.M.; resources, M.E.O., M.M. and A.A.; data curation, M.E.O.; writing original draft preparation, M.E.O., M.G. and S.E.; writing—review and editing, M.E.O., M.G. and S.E.; supervision, M.E.O.; project administration, M.E.O. All authors have read and agreed to the published version of the manuscript.

**Funding:** This research work was funded by the Deputyship for Research & Innovation, Ministry of Education in Saudi Arabia, under the project number IFPRC–082–123–2020.

**Institutional Review Board Statement:** Not applicable.

**Informed Consent Statement:** Not applicable.

**Data Availability Statement:** All data are provided as tables and figures.

**Acknowledgments:** "The authors extend their appreciation to the Deputyship for Research & Innovation, Ministry of Education in Saudi Arabia, for funding this research work through the project number IFPRC–082–123–2020" and King Abdulaziz University, DSR, Jeddah, Saudi Arabia.

**Conflicts of Interest:** The authors declare that they have no known competing financial interest or personal relationships that could have appeared to influence the work reported in this paper.

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
