# Peer review of "Groundwater Suitability for Drinking and Irrigation Using Water Quality Indices and Multivariate Modeling in Makkah Al-Mukarramah Province, Saudi Arabia"

_water, doi:10.3390/w14030483_

Round 1
Reviewer 1 Report
Manuscript title: Evaluation of Groundwater Quality and Suitability for Drinking and Irrigation using Water Quality Indices, and Multivariate Modeling in Makkah Al-Mukarramah province, Saudi Arabia
This study evaluated the groundwater quality in Makkah Al-Mukarramah Province, Saudi Arabia by testing fifty-nine groundwater wells. This study can provide guidance for future water quality assessment. However, there are some expression and format errors in this manuscript, and the pictures are not satisfactory. Meanwhile, important details are missing from the methods section that must be added. After careful revision, this manuscript can be accepted for publication.
Abstract: The narrative is cumbersome and the innovations and importance of this manuscript are not highlighted.
Figures: Need further optimization to improve aesthetics.
- Line 48-50: A reference citation is needed here to support the claims made about the water concerns.
- Line 72-74: A reference citation is needed here to support the claims made about the geochemical characteristics of groundwater.
- Line 123: The ‘WQIS’ here is different from ‘WQIs’ elsewhere or not.
- Line 141: Figure 1(b) contains format errors.
- Line 177: Please provide detailed detection methods and quality control results in 2.2 Samples collection and analytical procedures.
- Line 492: The coefficients values of R2 of IWQs and SAR in Val are only 0.848 and 0.861, which represent the fitting degree is low. Whether this will affect the prediction results?
- Line 537: The IWQI figure contains an error. ‘DWQI’ should be modified to ‘IWQI’.
Author Response
Reviewer 1
Manuscript title: Evaluation of Groundwater Quality and Suitability for Drinking and Irrigation using Water Quality Indices, and Multivariate Modeling in Makkah Al-Mukarramah province, Saudi Arabia
This study evaluated the groundwater quality in Makkah Al-Mukarramah Province, Saudi Arabia by testing fifty-nine groundwater wells. This study can provide guidance for future water quality assessment. However, there are some expression and format errors in this manuscript, and the pictures are not satisfactory. Meanwhile, important details are missing from the methods section that must be added. After careful revision, this manuscript can be accepted for publication.
We greatly appreciate your critical observations as well as your constructive and helpful comments. We hope that we could address your questions/comments by the explanations and revisions made in the manuscript. We believe that the manuscript is substantially improved after making the suggested revisions.
Abstract: The narrative is cumbersome and the innovations and importance of this manuscript are not highlighted.
Response: Many thanks for this comment. The abstract was modified to be concise sentences. The innovation and importance of the manuscript were highlighted using track changes in the abstract section as you suggested (Line 15 to 18 and line 46 to 49).
Figures: Need further optimization to improve aesthetics.
Response: We greatly appreciate your critical observations. The figures in all manuscript were optimized as your suggestion.
- Line 48-50: A reference citation is needed here to support the claims made about the water concerns.
Response: Many thanks for this comment. References citations were added as your suggestion (Line 57).
- Line 72-74: A reference citation is needed here to support the claims made about the geochemical characteristics of groundwater.
Response: Many thanks for this comment. References citations were added as your suggestion (Line 86).
- Line 123: The ‘WQIS’ here is different from ‘WQIs’ elsewhere or not.
Response: We greatly appreciate your critical observations. The "WQIS" was corrected to WQIs (Line 131).
- Line 141: Figure 1(b) contains format errors.
Response: We greatly appreciate your critical observations. The format of Figure 1(b) was corrected and was changed to (c) (Line 151 to 155).
- Line 177: Please provide detailed detection methods and quality control results in 2.2 Samples collection and analytical procedures.
Response: Many thanks for this comment. The detection methods and quality control results were presented in details as your suggestion (Line 214 to 234).
- Line 492: The coefficients values of R2 of IWQI and SAR in Val are only 0.848 and 0.861, which represent the fitting degree is low. Whether this will affect the prediction results?
Response: Many thanks for this comment. The performance of this model cannot increase the value of R2 for IWQI and SAR. In future studies, we will focus to use different machine learning models to test and improve the prediction results of IWQI and SAR.
- Line 537: The IWQI figure contains an error. ‘DWQI’ should be modified to ‘IWQI’.
Response: We greatly appreciate your critical observations. In Figure 10, the DWQI was modified to IWQI.

Reviewer 2 Report
Evaluation of Groundwater Quality and Suitability for Drinking and Irrigation using Water Quality Indices, and Multivariate Modeling in Makkah Al-Mukarramah Province, Saudi Arabia
The authors of this manuscript are based on the hydrochemical analysis of water sampled from Fifty-nine groundwater wells in arid and semi-arid regions of Makkah Al-Mukarramah Province, Saudi Arabia. They tried to determine hydrochemical facies, and hydrochemical processes leading to these facies. According to these analyses, they use water quality indices to assess potability and irrigation uses. With some multivariate statistical analysis (CA, PCA) and PLSR, they conclude that the PLSR models are more efficient in estimating the DWQI, IWQI, TDS, SAR, PS, and RSC, with the highest R2, lowest RMSE and MAD values, and highest slope values for both calibrating and validating data sets.
The manuscript has low language mistakes in style and spelling. It is well written with an easy style to read.
Below are some non-exhaustive remarks for the improvement of this manuscript:
- Page 1 line 18: spacing “parameter using”
- 2: - The two parts of figure 2 must be assigned as (a) and (b).
- In the geological map, Jurassic is not shown in the figure. If so, you must eliminate Jurassic legend.
- In the DEM map, generally, we use red color for the highest level and the blue color to the lowest level. Correct it!
- Page 5 line 152 – 155: eliminate repeated sentence.
- Page 6 line 190: clear “and” at the beginning of the line.
- Page 7. How do you calculate the constant of proportionality k in formula 3. You are invited to show this.
- In table 3, you must interfere the spiling of unit weight wi and arithmetic weight Wi.
- Page 11 line 334: Correct spelling “aquifer matrix”
- In the Gibbs plot, the dominance spread of most points is in the evaporation domain. Only part of them can be observed in the weathering and rock-water interaction field!! The high content of chloride and sulfates support the evaporation against interaction. Then weathering and rock-water interaction were not the first significant processes regulating the groundwater chemistry as you announced.
- Page 12 line 360: What mean KR and SSP? may be PS , SAR !!
- Page 14 Line 410 : Puntamkar et al. 1988
- Page 15 line 436: “purposewithno” must be separated.
Line 425 correct date of Subramani 2005
- Page 16 line 468: invistigated area as aresult (correct and separate the letter a)
I do not agree with the conclusion that there is deterioration in groundwater quality in the investigated area as a result of the weathering process, rock-water interaction, and saltwater intrusion. As the region is highly arid, it is evaporation as shown in Gibbs' diagram that takes over with the marine invasion. Weathering and interaction with the storage rock can be classified second as the process involved in this case.
- Page 18 line 508 : Mustapha et al., 2012
- Page 18 line 518 : The RMSE value for DWQI in the Val dataset is 10.030, not 7.686
- Some references are not cited in the text such as Abbasnia et al. 2018a ; Gupta and Gupta 2015 ; Suarez et al., 2006.
- In the reference list, there are letters a, b, c… !! What is their significance? I have searched many articles in the WATER journal, there are nowhere letters in the bibliographic lists.
- What about Zam Zam well, was it also sampled since it is part of the study area ??

Author Response
Reviewer 2
Manuscript title: Evaluation of Groundwater Quality and Suitability for Drinking and Irrigation using Water Quality Indices, and Multivariate Modeling in Makkah Al-Mukarramah Province, Saudi Arabia
The authors of this manuscript are based on the hydrochemical analysis of water sampled from Fifty-nine groundwater wells in arid and semi-arid regions of Makkah Al-Mukarramah Province, Saudi Arabia. They tried to determine hydrochemical facies, and hydrochemical processes leading to these facies. According to these analyses, they use water quality indices to assess possibility and irrigation uses. With some multivariate statistical analysis (CA, PCA) and PLSR, they conclude that the PLSR models are more efficient in estimating the DWQI, IWQI, TDS, SAR, PS, and RSC, with the highest R2, lowest RMSE and MAD values, and highest slope values for both calibrating and validating data sets.
The manuscript has low language mistakes in style and spelling. It is well written with an easy style to read.
We greatly appreciate your critical observations as well as your constructive and helpful comments. We hope that we could address your questions/comments by the explanations and revisions made in the manuscript. We believe that the manuscript is substantially improved after making the suggested revisions.
Below are some non-exhaustive remarks for the improvement of this manuscript:
- Page 1 line 18: spacing “parameter using”
Response: We greatly appreciate your critical observations. "parameter using" was separated by adding spacing (Line 19).
- The two parts of figure 2 must be assigned as (a) and (b).
Response: Many thanks for this comment. Figure 2 was assigned as (a) and (b) as your suggestion (Line 170).
In the geological map, Jurassic is not shown in the figure. If so, you must eliminate Jurassic legend.
Response: We greatly appreciate your critical observations. In the geological map, Jurassic is found towards the west but it is very small area about 2 km x 0.75 km, so it did not show in the map.
In the DEM map, generally, we use red color for the highest level and the blue color to the lowest level. Correct it!
Response: Many thanks for this comment. The DEM map was modified according to your suggestion.
- Page 5 line 152 – 155: eliminate repeated sentence.
Response: We greatly appreciate your critical observations. The repeated sentence was deleted (Line 178 to 181).
- Page 6 line 190: clear “and” at the beginning of the line.
Response: We greatly appreciate your critical observations. The word "and" at the beginning of the line was deleted (Line 222).
- Page 7. How do you calculate the constant of proportionality k in formula 3. You are invited to show this.
Response: Many thanks for this comment. The constant of proportionality k was calculated using formula 5 which, was added in the manuscript under indexing method section (Line 262 to 263).
- In table 3, you must interfere the spiling of unit weight wi and arithmetic weight Wi.
Response: Many thanks for this comment. The unit weight wi and relative weight Wi were written as your suggestion.
- Page 11 line 334: Correct spelling “aquifer matrix”
Response: We greatly appreciate your critical observations. The spelling of aquifer matrix was corrected (Line 391).
- In the Gibbs plot, the dominance spread of most points is in the evaporation domain. Only part of them can be observed in the weathering and rock-water interaction field!! The high content of chloride and sulfates support the evaporation against interaction. Then weathering and rock-water interaction were not the first significant processes regulating the groundwater chemistry as you announced.
Response: Many thanks for this comment. Evaporation process was added as a significant process regulating the groundwater chemistry and quality (Line 395).
- Page 12 line 360: What mean KR and SSP? may be PS , SAR !!
Response: We greatly appreciate your critical observations. These indices were corrected to PS, SAR, and RSC (Line 417).
- Page 14 Line 410 : Puntamkar et al. 1988
Response: We greatly appreciate your critical observations. This reference citation was modified as the number.
- Page 15 line 436: “purposewithno” must be separated.
Response: We greatly appreciate your critical observations. These words were corrected and separated by adding a spacing "purpose with no" (Line 493).
Line 425 correct date of Subramani 2005
Response: We greatly appreciate your critical observations. The date of the reference citation was modified as the number.
- Page 16 line 468: invistigated area as aresult (correct and separate the letter a)
Response: Many thanks for this comment. These words were corrected and separate the letter a (Line 527).
I do not agree with the conclusion that there is deterioration in groundwater quality in the investigated area as a result of the weathering process, rock-water interaction, and saltwater intrusion. As the region is highly arid, it is evaporation as shown in Gibbs' diagram that takes over with the marine invasion. Weathering and interaction with the storage rock can be classified second as the process involved in this case.
Response: Many thanks for this comment. According to geochemical properties of the groundwater, evaporation and salt-water intrusion were the predominantly factors controlling the quality of groundwater in the region (Line 527 to 531).
- Page 18 line 508 : Mustapha et al., 2012
Response: We greatly appreciate your critical observations. The date of the reference was modified as the number.
- Page 18 line 518 : The RMSE value for DWQI in the Val dataset is 10.030, not 7.686
Response: We greatly appreciate your critical observations. The RMSE value for DWQI in the Val dataset was modified to 10.030 (Line 575).
- Some references are not cited in the text such as Abbasnia et al. 2018a ; Gupta and Gupta 2015 ; Suarez et al., 2006.
Response: We greatly appreciate your critical observations. These references were cited in the text (Line 273, line 303, and line 484).
- In the reference list, there are letters a, b, c… !! What is their significance? I have searched many articles in the WATER journal, there are nowhere letters in the bibliographic lists.
Response: We greatly appreciate your critical observations. The letters a, b, c… were deleted from the references list.
- What about Zam Zam well, was it also sampled since it is part of the study area ??
Response: Many thanks for this comment. Zam Zam well was out of the catchment area.

Reviewer 3 Report
This manuscript well-structured and requires minor revision to be accepted for publication. Manuscript is about groundwater quality, its suitability for drinking and irrigation using water quality indices and multivariate analysis, in an arid area. General and specific comments are given below:
Title: It would be better to change the title for appropriate wording and sentence to “Groundwater suitability for drinking and irrigation using water quality indices and multivariate modeling in Makkah Al-Mukarramah, Saudi Arabia”.
Abstract: Make it concise. For example, lines 23 to 28, you can write “Drinking water quality-index (DWQI) has indicated that only 5% of wells were categorized under good to excellent for drinking. While majority (95%) were poor to unsuitable for drinking, require appropriate treatment”. You can concise whole abstract in similar way.
Introduction: It is fine and written scientifically. But, avoid phrases and sentences with similar meanings again and again.
Please write about the groundwater problems in arid area and why do you select Wadi Fatimah.
Methods: In samples collection and analytical procedures, it will be interesting to know the depth of groundwater wells, as it plays important role in water chemistry.
Rearrange Figure 1, for better understanding.
Table 1 is duplication of what is written in the text. You should add this information in the text. There is no need of Table 1.
Table 2 Units are missing in first column of parameters. Third, fourth, and fifth column are arbitrarily arranged. Please write 2 to 3 digit after point, for example, 0.415, and follow the same number of digits in all three columns.
Results and discussion: Table 3 again follow synchronization. Two digits after the point would be better. Example, 0.00. Similarly, it should be followed in text as well.
Conclusions are written following the results of this study. Make sure it presents all important findings.

Author Response
Reviewer 3
This manuscript well-structured and requires minor revision to be accepted for publication. Manuscript is about groundwater quality, its suitability for drinking and irrigation using water quality indices and multivariate analysis, in an arid area. General and specific comments are given below:
Title: It would be better to change the title for appropriate wording and sentence to “Groundwater suitability for drinking and irrigation using water quality indices and multivariate modeling in Makkah Al-Mukarramah, Saudi Arabia”.
Response: Many thanks for this comment. The title of the manuscript was changed to “Groundwater suitability for drinking and irrigation using water quality indices and multivariate modeling in Makkah Al-Mukarramah province, Saudi Arabia” as your suggestion (Line 1 to 4), except for the word " province ", it is important to put it in the tittle.
Abstract: Make it concise. For example, lines 23 to 28, you can write “Drinking water quality-index (DWQI) has indicated that only 5% of wells were categorized under good to excellent for drinking. While majority (95%) was poor to unsuitable for drinking, require appropriate treatment”. You can concise whole abstract in similar way.
Response: Many thanks for this comment. The abstract section was modified and improved as your suggestion (Line 23 to 28).
Introduction: It is fine and written scientifically. But, avoid phrases and sentences with similar meanings again and again.
Response: Many thanks for this comment. The introduction section was improved and the similar sentences were deleted as your suggestion (Line 75 to 78) and also (Line 178 to 181).
Please write about the groundwater problems in arid area and why do you select Wadi Fatimah.
Response: Many thanks for this comment. The groundwater problems in arid area were presented and the selection of Wadi Fatimah for this study was introduced as your suggestion (Line 57 to 63).
Methods: In samples collection and analytical procedures, it will be interesting to know the depth of groundwater wells, as it plays important role in water chemistry.
Response: Many thanks for this comment. The depth of groundwater wells was presented under the materials and methods section (Line 196 to 198).
Rearrange Figure 1, for better understanding.
Response: Many thanks for this comment. Figure 1 was rearranged for better understanding as your suggestion.
Table 1 is duplication of what is written in the text. You should add this information in the text. There is no need of Table 1.
Response: Many thanks for this comment. Table 1 was deleted and the information about the indices was added in the text under material and method as your suggestion (line 288 to 303).
Table 2 Units are missing in first column of parameters. Third, fourth, and fifth column are arbitrarily arranged. Please write 2 to 3 digit after point, for example, 0.415, and follow the same number of digits in all three columns.
Response: Many thanks for this comment. The units in the first column were added. Also, third, fourth, and fifth column were arranged as your suggestion. In addition, the number of digits in all three columns was written in 3 digits after point, as your suggestion (Line 270).
Results and discussion: Table 3 again follows synchronization. Two digits after the point would be better. Example, 0.00. Similarly, it should be followed in text as well.
Response: Many thanks for this comment. Two digits after the point were done in the Table and text as your suggestion (Line 356 to 361).
Conclusions are written following the results of this study. Make sure it presents all important findings.
Response: Many thanks for this comment. All important findings in this study were highlighted by using track changes in the conclusion section as your suggestion.

Round 2
Reviewer 1 Report
This is an interesting and valuable work. We believe that this manuscript is suitable for publication after authors' careful revision.